# Comparing human and model-based forecasts of COVID-19 in Germany and Poland

**Nikos I. Bosse**[1,2]*, **Sam Abbott**[1,2], **Johannes Bracher**[3], **Habakuk Hain**[4], **Billy J. Quilty**[1,2], **Mark Jit**[1,2], **Centre for the Mathematical Modelling of Infectious Diseases COVID-19 Working Group**[1,2], **Edwin van Leeuwen**[1,5], **Anne Cori**[6], **Sebastian Funk**[1,2]

**1** Department of Infectious Disease Epidemiology, London School of Hygiene & Tropical Medicine, London, United Kingdom, **2** Centre for the Mathematical Modelling of Infectious Diseases (members of the CMMID COVID-19 working group are listed in S1 Acknowledgements), London, United Kingdom, **3** Institute of Economic Theory and Statistics, Karlsruhe Institute of Technology, Karlsruhe, Germany, **4** Max Planck Institute for Multidisciplinary Sciences, Göttingen, Germany, **5** UK Health Security Agency, London, United Kingdom, **6** MRC Centre for Outbreak Analysis and Modelling, Department of Infectious Disease Epidemiology, School of Public Health, Imperial College London, London, United Kingdom

* nikos.bosse@lshtm.ac.uk

**Data Availability Statement:** All code is available under https://github.com/epiforecasts/covid. german.forecasts.

## Abstract

Forecasts based on epidemiological modelling have played an important role in shaping public policy throughout the COVID-19 pandemic. This modelling combines knowledge about infectious disease dynamics with the subjective opinion of the researcher who develops and refines the model and often also adjusts model outputs. Developing a forecast model is difficult, resource- and time-consuming. It is therefore worth asking what modelling is able to add beyond the subjective opinion of the researcher alone. To investigate this, we analysed different real-time forecasts of cases of and deaths from COVID-19 in Germany and Poland over a 1-4 week horizon submitted to the German and Polish Forecast Hub. We compared crowd forecasts elicited from researchers and volunteers, against a) forecasts from two semi-mechanistic models based on common epidemiological assumptions and b) the ensemble of all other models submitted to the Forecast Hub. We found crowd forecasts, despite being overconfident, to outperform all other methods across all forecast horizons when forecasting cases (weighted interval score relative to the Hub ensemble 2 weeks ahead: 0.89). Forecasts based on computational models performed comparably better when predicting deaths (rel. WIS 1.26), suggesting that epidemiological modelling and human judgement can complement each other in important ways.

## Author Summary

Mathematical models of COVID-19 have played a key role in informing governments across the world. While mathematical models are informed by our knowledge of infectious disease dynamics, they are ultimately developed and iteratively adjusted by the researchers and shaped by their subjective opinions. To investigate what modelling is able to add beyond the subjective opinion of the researcher alone, we compared human forecasts with model-based predictions of COVID-19 cases and deaths submitted to the so-

**Funding:** NIB received funding from the National Institute for Health Research (NIHR) Health Protection Research Unit (grant code NIHR200908, https://www.nihr.ac.uk/). SA's work was funded by the Wellcome Trust (grant: 210758/Z/18/Z, https://wellcome.org/). The work of JB was supported by the Helmholtz Foundation (https://www.helmholtz.de/) via the SIMCARD Information and Data Science Pilot Project. The work of BJQ was partly funded by the National Institute for Health Research (NIHR, https://www.nihr.ac.uk/) (16/137/109 & 16/136/46) using UK aid from the UK Government to support global health research. The views expressed in this publication are those of the author(s) and not necessarily those of the NIHR or the UK Department of Health and Social Care. BJQ is also supported in part by a grant from the Bill and Melinda Gates Foundation (OPP1139859, https://www.gatesfoundation.org/). MJ and EvL acknowledge funding by the National Institute for Health Research (NIHR) Health Protection Research Unit (HPRU) in Modelling and Health Economics (grant number NIHR200908, https://www.nihr.ac.uk/) and the European Union's Horizon 2020 research and innovation programme - project EpiPose (101003688, https://ec.europa.eu/programmes/horizon2020/). AC acknowledges funding by the NIHR, the Sergei Brin foundation, USAID (https://www.usaid.gov/), and the Academy of Medical Sciences (https://acmedsci.ac.uk/). SF's work was supported by the Wellcome Trust (grant: 210758/Z/18/Z, https://wellcome.org/). The funders had no role in study design, data collection and analysis, decision to publish, or preparation of the manuscript.

**Competing interests:** The authors have declared that no competing interests exist.

called German/Polish Forecast Hub (which collates a variety of models from a range of teams). We found that our human forecasts consistently outperformed an aggregate of all available model-based forecasts when predicting cases, but not when predicting deaths. Our findings suggest that human insight may be most valuable when forecasting highly uncertain quantities, which depend on many factors that are hard to model using equations, while mathematical models may be most useful in settings like predicting deaths, where leading indicators with a clear connection to the target variable are available. This potentially has very relevant policy implications, as agencies informing policy-makers could benefit from routinely eliciting human forecasts in addition to model-based predictions to inform policies.

## Introduction

Infectious disease modelling has a long tradition and has helped inform public health decisions both through scenario modelling, as well as actual forecasts of (among others) influenza [e.g. 1,2–4], dengue fever [e.g. 5,6,7], ebola [e.g. 8,9], chikungunya [e.g. 10,11] and now COVID-19 [e.g. 12,13–17]. Applications of epidemiological models differ in the way they make statements about the future. Forecasts aim to predict the future as it will occur, while scenario modelling and projections aim to represent what the future could look like under certain scenario assumptions or if conditions stayed the same as they were in the past. Forecasts can be judged by comparing them against observed data. Since it is much harder to fairly assess the accuracy and usefulness of projections and scenario modelling in the same way, this work focuses on forecasts, which represent only a subset of all epidemiological modelling.

Since March 2020, forecasts of COVID-19 from multiple teams have been collected, aggregated and compared by Forecast Hubs such as the US Forecast Hub [13, 14], the German and Polish Forecast Hub [15, 16] and the European Forecast Hub [17]. Often, different individual forecasts are combined into a single forecast, e.g. by taking the mean or median of all forecasts. These ensemble forecasts usually tend to perform better and more consistently than individual forecasts (see e.g. [6]; [18]).

Individual computational models usually rely to varying degrees on mechanistic assumptions about infectious disease dynamics (such as SIR-type compartmental models that aim to represent how individuals move from being susceptible to infected and then recovered or dead). Some are more statistical in nature (such as time series models that detect statistical patterns without explicitly modelling disease dynamics). How exactly such a mathematical or computational model is constructed and which assumptions are made depends on subjective opinion and judgement of the researcher who develops and refines the model. Models are commonly adjusted and improved based on whether the model output looks plausible to the researchers involved.

The process of model construction and refinement is laborious and time-consuming, and it is therefore worth asking what modelling can add beyond the subjective judgment of the researcher alone. In this work, we ask this question specifically in the context of predictive performance, and set aside other advantages of epidemiological modelling (such as reproducibility or the ability to obtain a deeper fundamental understanding of how diseases spread). One natural way to do this is to compare the predictive performance of forecasts based on computational models ("model-based forecasts") against forecasts made by individual humans without explicit use of a computer model ("direct human forecasts") or a combination of multiple such forecasts ("crowd forecasts").

Previous work has examined such direct human forecasts in various contexts, such as geopolitics [19, 20], meta-science [21, 22], sports [23] and epidemiology [11, 24, 25]. Several prediction platforms [26–28] and prediction markets [29] have been created to collate expert and non-expert predictions. However, with the notable exception of [11], these forecasts were not designed to be evaluated alongside model-based forecasts and usually follow their own (often binary) prediction formats. Direct human forecasts may be able to take into account insights and relationships between variables which are hard to specify using epidemiological models. However, it is not entirely clear in which situations human forecasts perform well or badly. For example, [11] found that humans could outperform computer models at predicting the 2014/15 and 2015/16 flu season in the US, a setting where the disease was well known and information about previous seasons was available. However, humans tended to do slightly worse at predicting the 2014/15 outbreak of chikungunya in the Americas, a disease previously largely unobserved and unknown in these regions at the time.

In this study, we analyse the performance of direct human forecasts relative to model-based forecasts and discuss the added benefit of epidemiological modelling over human judgement alone. As a case study, we use different forecasts, involving varying degrees of human intervention, which we submitted in real time to the German and Polish Forecast Hub. In contrast to [11] we elicited not only point predictions, but full predictive distributions ("probabilistic forecasts", see e.g. [30]) from participants. This allows us to compare not only predictive accuracy, but also how well human forecasters and model-based forecasts were able to quantify forecast uncertainty.

## Methods

### Ethics statement

This study has been approved by the London School of Hygiene & Tropical Medicine Research Ethics Committee (reference number 22290). Consent from participants was obtained in written form.

### Overview

We created and submitted the following forecasts to the German and Polish Forecast Hub: 1) a direct human forecast (henceforth called "crowd forecast"), elicited from participants through a web application [31] and 2) two semi-mechanistic model-based forecasts ("renewal model" and "convolution model") informed by basic assumptions about COVID-19 epidemiology. While the two semi-mechanistic forecasts were necessarily shaped by our implicit assumptions and decisions, they were designed such as to minimise the amount of human intervention involved. For example, we refrained from adjusting model outputs or refining the models based on past performance. Forecasts were created in real time over a period of 21 weeks from October 12th 2020 until March 1st 2021 and submitted to the German and Polish Forecast hub [15, 16]. All code and tools necessary to generate the forecasts and make a forecast submission are available in the `covid.german.forecasts` R package [32]. This repository also contains a record of all forecasts submitted to the German and Polish Forecast Hub. Forecasts were evaluated using a variety of scoring metrics and compared among each other and against an ensemble of all other models submitted to the German and Polish Forecast Hub.

### Forecast targets and interaction with the German and Polish Forecast Hub

The German and Polish Forecast Hub (now mostly merged into the European Forecast Hub [17]) elicits predictions for various COVID-19 related forecast targets from different research

groups every week. Forecasts had to be made every Monday (with submissions allowed until Tuesday 3pm) and were permitted to use any data that was available by Monday 11.59pm. We submitted forecasts for incident and cumulative weekly reported numbers of cases of and deaths from COVID-19 on a national level in Germany and Poland over a one to four week forecast horizon. Forecasts were submitted on Mondays, but weeks were defined as ending on a Saturday (and starting on Sunday), meaning that forecast horizons were in fact 5, 12, 19 and 26 days. Submissions were required in a quantile-based format with 23 quantiles of each output measure at levels 0.01, 0.025, 0.05, 0.10, 0.15,. . ., 0.95, 0.975, 0.99. Forecasts submitted to the Forecast Hub were combined into different ensembles every week, with the median ensemble (i.e., the $\alpha$-quantile of the ensemble is given by the median of all submitted $\alpha$-quantiles) being the default ensemble shown on all official Forecast hub visualisations (https:// kitmetricslab.github.io/forecasthub/forecast).

Data on daily reported test positive cases and deaths linked to COVID-19 were provided by the organisers of the German and Polish Forecast hub. Until December 14th, 2020, these data were sourced from the European Centre for Disease Control [33]. After ECDC stopped publishing daily data, observations were sourced from the Robert Koch Institute (RKI) and the Polish Ministry of Health for the remainder of the submission period [34]. These data are subject to reporting artefacts, (such as for example delayed case reporting in Poland on the 24th November, [35]), changes in reporting over time, and variation in testing regimes (for example in Germany from the 11th of November on, [36]). The ECDC data as well as the data published by the Polish Ministry of Health were also subject to data revisions, although most of them (with a notable exception of a data update for October 12 2020 in Germany) only affected daily, not weekly data (see S7 and S8 Figs).

## Crowd forecasts

Our crowd forecasts were created as an ensemble of forecasts made by individual participants every week through a web application (https://cmmid-lshtm.shinyapps.io/crowd-forecast/). Weekly forecasts had to be submitted before Tuesday 12pm every week, but participants were asked to only use any information or data that was already available by Monday night. The application was built using the shiny and golem R packages [37, 38] and is available in the crowdforecastr R package [31]. To make a forecast in the application participants could select a predictive distribution (with the default being log-normal) to represent the probability that the forecasted quantity took certain values. Median and width of the uncertainty could be adjusted by either interacting with a figure showing their forecast or providing numerical values (see screenshot in S1 Fig). The default shown was a repetition of the last known observation with constant uncertainty around it computed as the standard deviation of the last four changes in weekly log observed forecasts (i.e. as $\sigma(log(value4) - log(value3), log(value3) - log(value2), . . .))$. A comparison of the crowd forecasts against the default baseline shown in the application is displayed in S25 Fig. Our interface also allowed participants to view past observations based on the hub data, as well as their forecasts, on a logarithmic scale and presented additional contextual COVID-19 data sourced from [39]. These data included, for example, notifications of both test positive COVID-19 cases and COVID-19 linked deaths and the number of COVID-19 tests conducted over time. From November 26 2020 on we displayed weekly small reports with a visualisation of past forecasts and scores on our website, epiforecasts.io.

Forecasts were stored in a Google Sheet and downloaded, cleaned and processed every week for submission to the Forecast Hub. If a forecaster had submitted multiple predictions for a single target, only the latest submission was kept. Information on the chosen distribution as well as the parameters for median and width were used to obtain the required set of 23

quantiles from that distribution. Forecasts from all forecasters were then aggregated using an unweighted quantile-wise mean (i.e., the $\alpha$-quantile of the ensemble is given by the mean of all submitted $\alpha$-quantiles). To avoid issues with users trying out the app and submitting a random forecast, we required that a forecaster needed to make a forecast for at least two targets for a given forecast in order to be included in the crowd forecast ensemble. On a few occasions we deleted forecasts that were clearly the result of a user or software error (such as for example forecasts that were zero everywhere).

Participants were recruited mostly within the Centre of Mathematical Modeling of Infectious Diseases at the London School of Hygiene & Tropical Medicine, but participants were also invited personally or via social media to submit predictions. Depending on whether they had a background in either statistics, forecasting or epidemiology, participants were asked to self-identify as 'experts' or 'non-experts'.

## Model-based forecasts

We used two Bayesian semi-mechanistic models from the `EpiNow2` R package (version 1.3.3) as our model-based forecasts [40]. The first of these models, here called "renewal model", used the renewal equation [41] to predict reported cases and deaths (see details in S1 Text). It estimated the effective reproduction number $R_t$ (the average number of people each person infected at time t is expected to infect in turn) and modelled future infections as a weighted sum of past infection multiplied by $R_t$. $R_t$ was assumed to stay constant beyond the forecast date, roughly corresponding to continuing the latest exponential trend in infections. On the 9th of November we altered the date when $R_t$ was assumed to be constant from two weeks prior to the date of the forecast to the forecast date, which we found to yield a more stable $R_t$ estimate. Reported case and death notifications were obtained by convolving predicted infections over data-based delay distributions [40, 42–44] to model the time between infection and report date. The renewal model was used to predict cases as well as deaths with forecasts being generated for each target separately. Death forecasts from the renewal model were therefore not informed by past cases. One submission of the renewal model on December 28th 2020 was delayed and therefore not included in the official Forecast hub ensemble.

The second model ("convolution model", see details in S1 Text). was only used to forecast deaths and was added later, starting December 7th 2020 (with the first forecast from December 7th suffering from a software bug and therefore disregarded in all further analyses). The convolution model was submitted, but never included in the official Forecast hub ensemble due to concerns that it could be too similar to the renewal model. The convolution model predicted deaths as a fraction of infected people who would die with some delay, by using a convolution of reported cases with a distribution that described the delay from case report to death and a scaling factor (the case-fatality ratio). Both the renewal and the convolution model used daily observations and assumed a negative binomial observation model with a multiplicative day-of-the-week effect [40].

Line list data used to inform the prior for the delay from symptom onset to test positive case report or death in the model-based forecasts was sourced from [45] with data available up to the 1st of August. All model fitting was done using Markov-chain Monte Carlo (MCMC) in stan [46] with each location and forecast target being fitted separately.

## Analysis

For the main analysis we focused mostly on two week ahead forecasts, as COVID-19 forecasts, especially for cases, were in the past found to have poor predictive performance beyond this horizon [15]. Forecasts for cases were scored using the full period from October 2020 until

March 2021. To ensure comparability between models, all death forecasts were scored using only the period from December 14th on, where all models including the convolution model were available. To ensure robustness of our results we conducted a sensitivity analysis where all forecasts (including cases) were scored only over the later period for which all forecasts were available (see S22 Fig and S8 and S9 Tables). Results remained broadly unchanged.

Forecasts were analysed using the following scoring metrics: The weighted interval score (WIS) [47], the absolute error, relative bias, and empirical coverage of the 50% and 90% prediction intervals. The WIS is a proper scoring rule [48], meaning that in expectation the score is optimised by reporting a predictive distribution that is identical to the true data-generating distribution. Forecasters are therefore incentivised to report their true belief about the future. The WIS can be understood as a generalisation of the absolute error to quantile-based forecasts (also meaning that smaller values are better) and can be decomposed into three separate penalties: forecast spread (i.e. uncertainty of forecasts), over-prediction and under-prediction. While the over- and under-prediction components of the WIS capture the amount of over-prediction and under-prediction in absolute terms, we also look at a relative tendency to make biased forecasts. The bias metric [9] we use captures how much probability mass of the forecast was above or below the true value (mapped to values between -1 and 1) and therefore represents a general tendency to over- or under-predict in relative terms. A value of -1 implies that all quantiles of the predictive distribution are below the observed value and a value of 1 that all quantiles are above the observed value. Empirical coverage is the percentage of observed values that fall inside a given prediction interval (e.g. how many observed values fall inside all 50% prediction intervals). Scoring metrics are explained in more detail in S1 Table. All scores were calculated using the scoringutils R package [49].

At all stages of the evaluation our forecasts were compared to the median ensemble of all *other* models submitted to the German and Polish Forecast Hub ("Hub ensemble"). This "Hub ensemble" was retrospectively computed and excludes all our models, leaving on average five ensemble member models (see S10 Table and S24 Fig). What we call "Hub ensemble" in this article therefore differs from the "official Hub ensemble" (here called "hub-ensemble-realised") which included crowd forecasts as well as renewal model forecasts. To enhance interpretability of scores we mainly report WIS relative to the Hub ensemble in the main text, i.e. we divided the average scores for a given model by the average score achieved by the Hub ensemble on the same set of forecasts (with values >1 implying worse and values <1 implying better performance than the Hub ensemble). In addition to comparing our forecasts against the hub ensemble excluding our models, we also assessed the impact of our forecasts on the performance of the forecasting hub by recalculating separate versions of the Hub ensemble with only some (or all) of our forecasts included. Versions that included either all of our models ("hub-ensemble-with-all") or only one of them ("hub-ensemble-with-X") were computed retrospectively.

## Results

### Crowd forecast participation

A total number of 32 participants submitted forecasts, 17 of those self-identified as 'expert' in either forecasting or epidemiology. The median number of forecasters for any given forecast target was 6, the minimum 2 and the maximum 10. The mean number of submissions from an individual forecaster was 4.7 but the median number was only one—most participants dropped out after their first submission. Only two participants submitted a forecast every single week, both of whom are authors on this study.

### Case forecasts

For cases, crowd forecasts had a lower mean weighted interval score (WIS, lower values indicate better performance) than both the renewal model and the Hub ensemble across all forecast horizons (Fig 1A) and locations (S5(A) Fig). For two week ahead forecasts, mean WIS relative to the Hub ensemble (= 1) was 0.89 for crowd forecasts and 1.40 for the renewal model (S2 Table). Across all forecasting approaches, locations and forecast horizons, the distribution of WIS values was very right-skewed, and average performance was heavily influenced by outliers (see Fig 2). Overall, low variance in forecast performance was closely linked with good

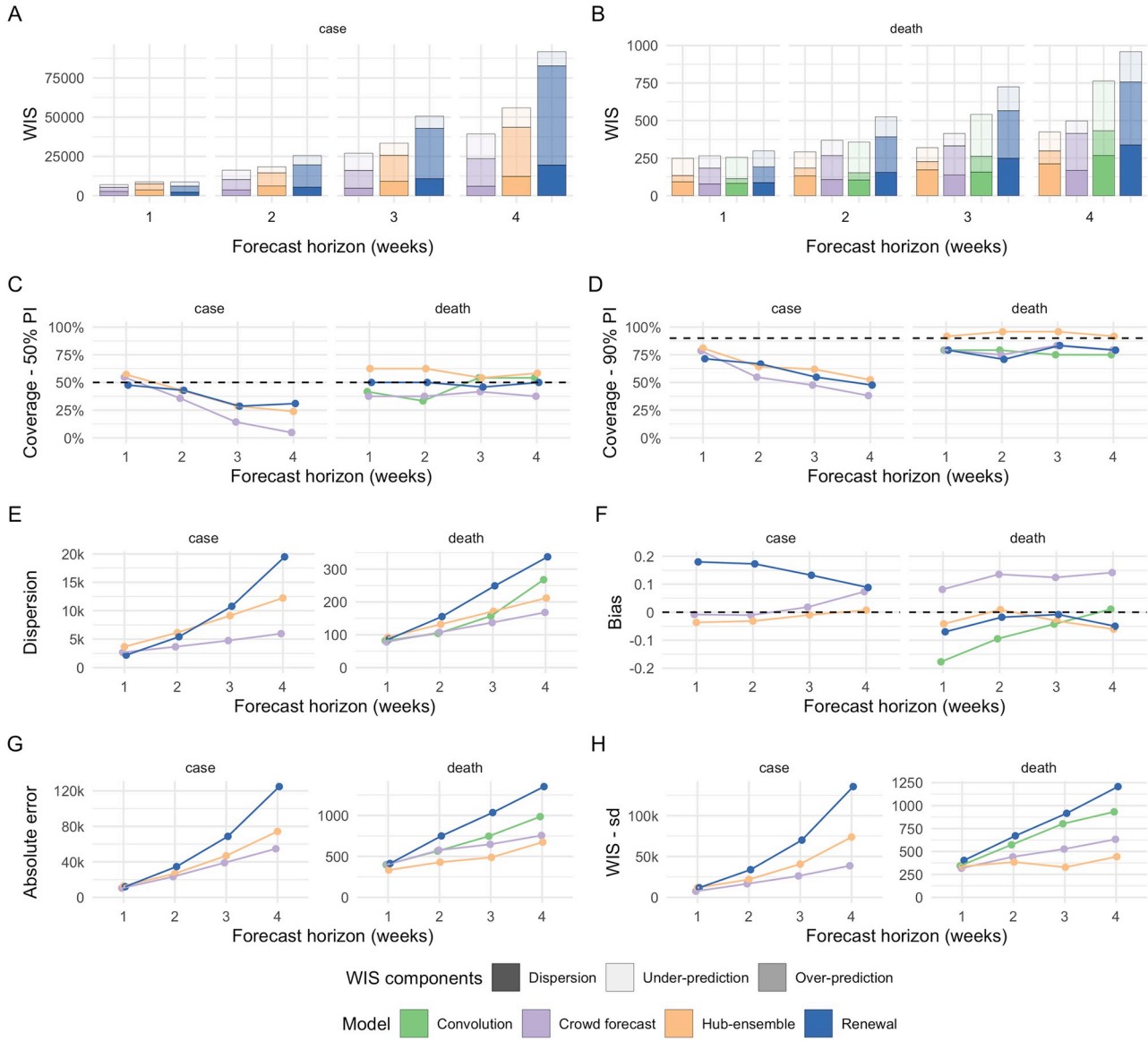

**Fig 1. Visualisation of aggregate performance metrics for forecasts one to four weeks into the future.** A, B: mean weighted interval score (WIS, lower indicates better performance) across horizons. WIS is decomposed into its components dispersion, over-prediction and under-prediction. C: Empirical coverage of the 50% prediction intervals (50% coverage is perfect). D: Empirical coverage of the 90% prediction intervals. E: Dispersion (same as in panel A, B). Higher values mean greater dispersion of the forecast and imply ceteris paribus a worse score. F: Bias, i.e. general (relative) tendency to over- or underpredict. Values are between -1 (complete under-prediction) and 1 (complete over-prediction) and 0 ideally. G: Absolute error of the median forecast (lower is better). H. Standard deviation of all WIS values for different horizons.

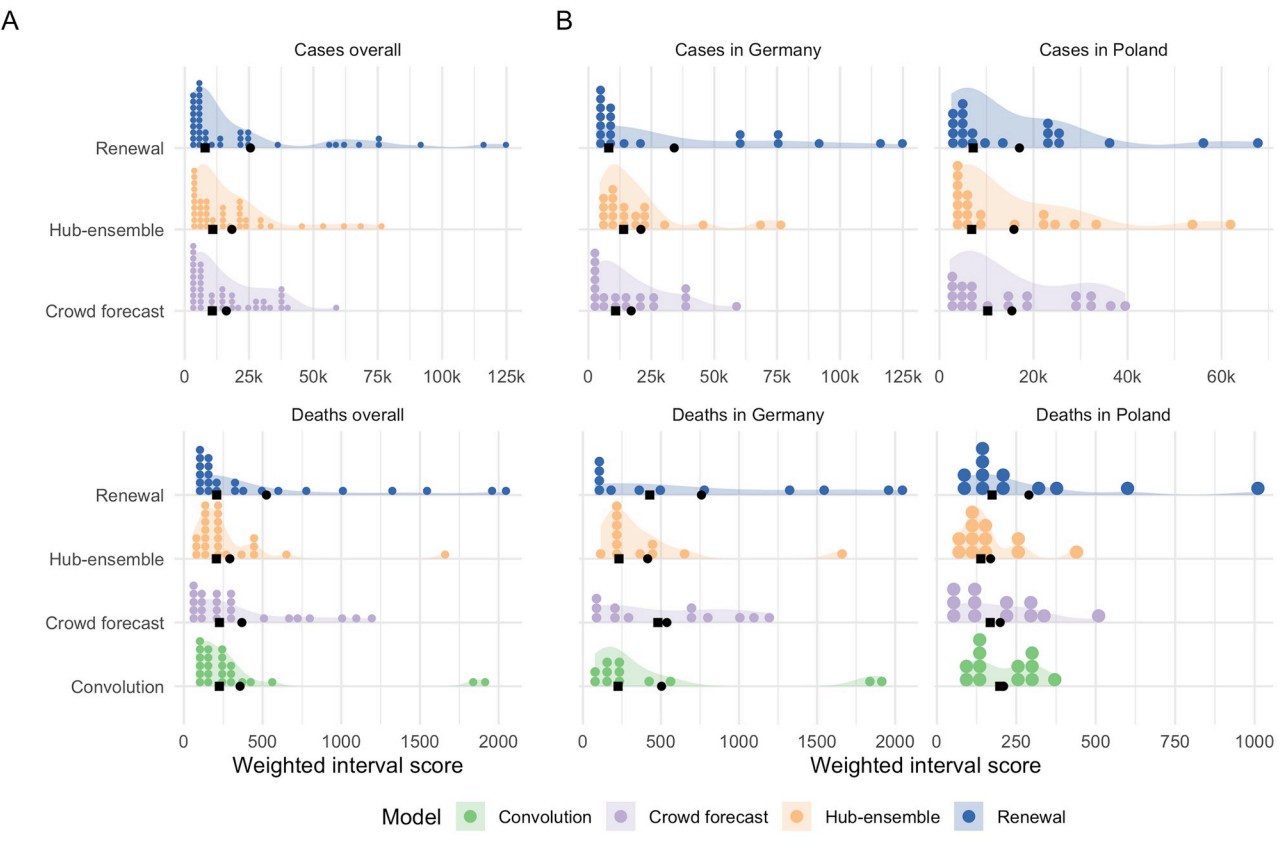

**Fig 2. Two week ahead forecasts and corresponding scores.** A, C: Visualisation of 50% prediction intervals of two week ahead forecasts against the reported values. Forecasts that were not scored (because there was no complete set of death forecasts available) are greyed out. B, D: Visualisation of corresponding WIS.

mean performance (Fig 1H and 1A), suggesting that the ability to avoid large errors was an important factor in determining overall performance. The impact of outlier values was especially pronounced for the renewal model, which had more outliers, as well as the highest standard deviation of WIS values (standard deviation of the WIS relative to the WIS sd of the Hub ensemble was 1.54 at the two weeks ahead horizon), while the ensemble of crowd forecasts (rel. WIS sd 0.76) and the Hub ensemble (= 1) showed more stable performance.

To varying degrees, all forecasts exhibited trend-following behaviour and were rarely able to predict a change in trend before it had happened. For example, all forecasts failed to predict the change in trend from increase to decrease that happened in November in Germany and severely overshot reported cases (Fig 3A). This was most striking for the renewal model, which extrapolated unconstrained exponential growth based on the recent past of observations. The Hub ensemble and the crowd forecast, which had both been under-predicting throughout October, also failed to predict the change in trend after cases peaked, but less severely so. Human forecasters, possibly aware of the semi-lockdown announced on November 2nd 2020 [50] and the change in the testing regime (with stricter test criteria) on November 11th 2020 [36], were fastest to adapt to the new trend, and the Hub ensemble slowest. In December, cases rose again in Germany, with all models under-predicting this growth to varying extents. As in October, the renewal model captured the phase of exponential growth in cases slightly better than other approaches, but again overshot when reported case numbers fell over Christmas.

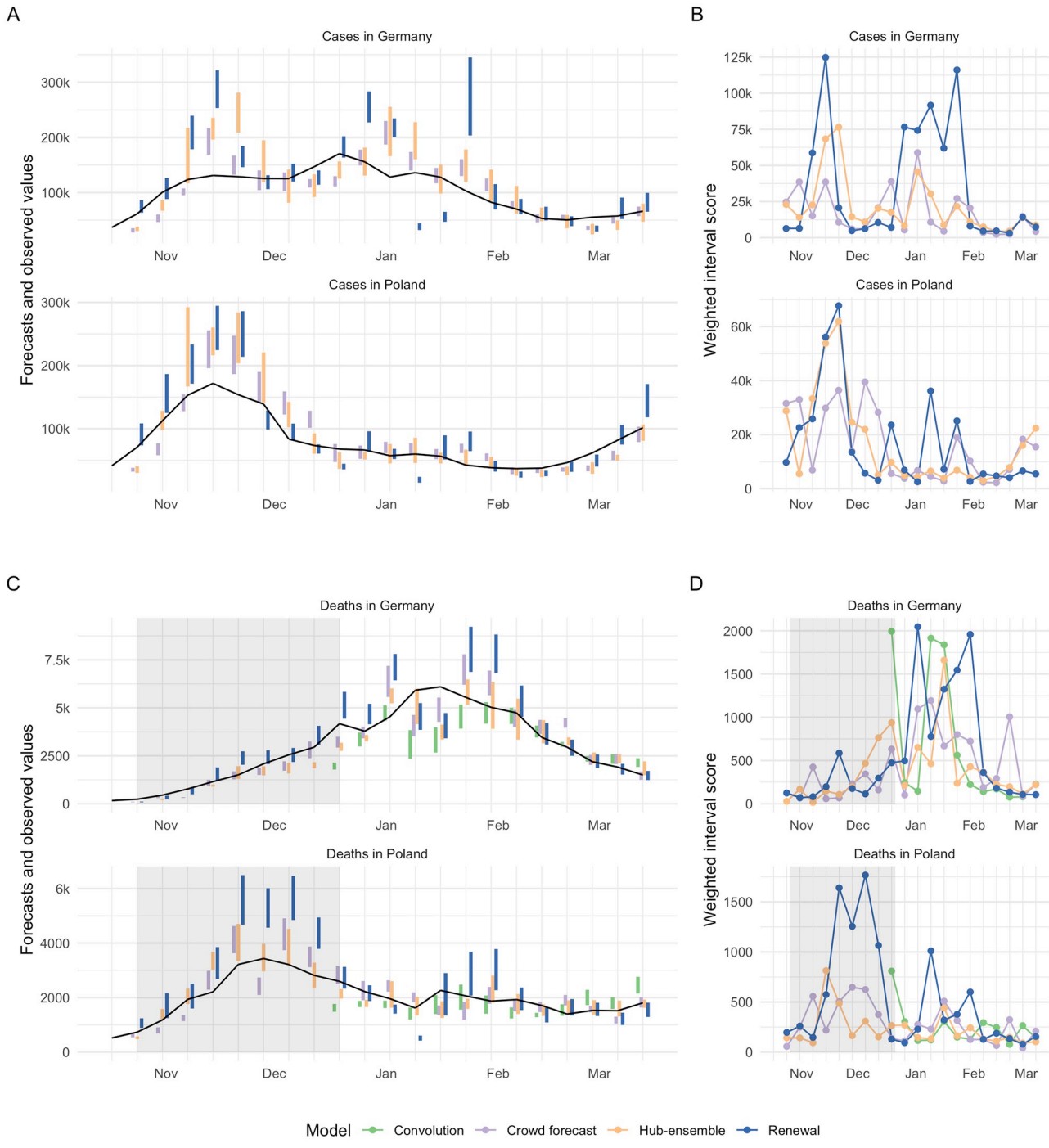

**Fig 3. Distribution of scores.** A: Distribution of weighted interval scores for two week ahead forecasts of the different models and forecast targets. Points denote single forecasts scores, while the shaded area shows an estimated probability density. B: Distribution of WIS separate by country. Black squares indicate median and black circles mean scores.

The large variance in predictions in January in Germany (severe under-prediction followed by severe over-prediction) may in part be caused by the fact that the renewal model operated on daily data and therefore was susceptible to fluctuations in daily reporting around Christmas that would not have influenced on weekly reporting. Similar trends in performance were

evident in Poland, with the crowd forecast quickest at adapting to the change in trend in November. In general, there were fewer large outlier forecasts in Poland and in particular the renewal model performed more in line with other forecasts there.

All forecasting approaches, including the Hub ensemble, were overconfident, i.e. they showed lower than nominal coverage (meaning that 50% (90%) prediction intervals generally covered less than 50% (90%) of the actually observed values) (Fig 1C and 1D). Coverage for all forecasts deteriorated with increasing forecast horizon, indicating that all forecasting approaches struggled to quantify uncertainty appropriately for case forecasts. This was especially an issue for crowd forecasts, which had markedly shorter prediction intervals (i.e., narrower and more confident predictive distributions) than other approaches (Fig 1E) and only showed a small increase in uncertainty across forecast horizons. The crowd forecasts prediction intervals were also noticeably narrower than the default baseline shown to forecasters in the application (see S25 Fig).

In spite of good performance in terms of the absolute error (Fig 1G), the narrow forecast intervals led to forecasts which were severely overconfident (covering only 36% and 55% of all observations with the 50% and 90% prediction intervals of all forecasts made at a two week forecast horizon, and only 5% and 38% four weeks ahead) (Fig 1C and 1D as well as S2 and S3 Tables). Despite worse performance in terms of absolute error (Fig 1G), the renewal model achieved better calibration (comparable to the Hub ensemble), as uncertainty increased rapidly across forecast horizons. The crowd forecasts, on the other hand, showed a smaller bias than the renewal model, but were overconfident.

The renewal model exhibited a noticeable tendency towards over-predicting reported cases across all horizons. The crowd forecast tended to over-predict at longer forecast horizons, whereas the Hub ensemble showed no systematic bias (Fig 1F). Regardless of a general relative tendency to over-predict, all forecasting approaches incurred larger absolute penalties from over- than from under-prediction (see decomposition of the WIS into absolute penalties for over-prediction, under-prediction and dispersion in Fig 1A and 1B, as well as S2 and S3 Tables).

Generally, trends in overall performance were broadly similar across locations (S4 and S5 Figs). Due to the differing population sizes and numbers of notifications in Germany and Poland absolute scores were difficult to compare directly. However, relative to the Hub ensemble, the crowd forecasts performed noticeably better in Germany than in Poland and the renewal model better in Poland than in Germany (S5(A), S5(G), S2 and S3 Figs).

## Death forecasts

For deaths, the Hub ensemble outperformed the crowd forecasts as well as our model-based approaches across all forecast horizons and locations (Fig 1B and S4(B) Fig). Relative WIS values for the models two weeks ahead were 1.22 (convolution model), 1.26 (crowd forecast), 1 (Hub ensemble) and 1.79 (renewal model). The crowd forecasts performed better than the renewal model across all forecast horizons and locations (Fig 1B and S4(B) Fig), and also better than the convolution model three and four weeks ahead. Poor performance of the renewal model, especially at longer horizons, indicates that an approach that does not know about past cases, but instead estimates and projects a separate $R_t$ trace from deaths, does not use the available information efficiently. The convolution model was able to outperform both the renewal model and the crowd forecasts at shorter forecast horizons (where the delay between cases and deaths means that future deaths are largely informed by present cases), but saw performance deteriorate at three and four weeks ahead (where case predictions from the renewal model were increasingly used to inform death predictions) (Fig 1B, S3 Table).

As past cases and hospitalisations can be used as predictors, predicting a change in trend may be easier for deaths than for cases. Even though all forecasts generally struggled with this, there were some instances where changing trends were well captured or even anticipated. In Poland, for example, the Hub ensemble was able to capture or even anticipate the peak in deaths in December quite well (whereas the renewal model and crowd forecast did not). The renewal model, which mostly exhibited trend-following behaviour, correctly predicted another increase in weekly deaths in mid-January (potentially based on changes in daily deaths, as the renewal model did not know about past cases). In Germany in early January, all models predicted a decrease in deaths two to three weeks before it actually happened. Predictions from the renewal model at that time were likely strongly influenced by an unexpected drop in reported deaths in December. The other forecasting approaches and in particular, the convolution model may have been affected by potentially under-reported case numbers around Christmas. When the decrease that all models had predicted to happen in early January failed to materialise, the renewal model and the crowd forecast noticeably over-corrected and over-predicted deaths in the following weeks, while the Hub ensemble, and to a slightly lesser degree, the convolution model were able to capture the downturn well when it finally happened at the end of January.

Death forecasts, generally, showed greater coverage of the 50% and 90% prediction intervals than case forecasts and no decrease in coverage across forecast horizons, indicating that it might be easier to appropriately quantify uncertainty for death forecasts. The Hub ensemble had the greatest coverage, with empirical coverage of the 50% and 90% prediction intervals exceeding 50%, and 90%, respectively, across all forecast horizons. Coverage for the crowd forecasts and our model-based approaches was generally lower than that of the Hub ensemble and mostly slightly lower than nominal coverage (Fig 1C and 1D). As for cases, the crowd forecast tended to have the narrowest prediction intervals and uncertainty increased most slowly across forecast horizons, and the renewal model forecasts generally were widest. The convolution model had relatively narrow prediction intervals for short forecast horizons, but had rapidly (and non-linearly) increasing uncertainty for longer forecast horizons, driven by increasing uncertainty in the underlying case forecasts.

For deaths, the ensemble of crowd forecasts had a consistent tendency to over-predict (see Fig 1F). The convolution model had a strong tendency to under-predict, with the magnitude of under-prediction steadily decreasing for longer forecast horizons. The renewal model (which over-predicted for cases) and the Hub ensemble slightly tended towards under-prediction. For deaths, absolute over- and under-prediction penalties were more in line with a general relative tendency to over- or under-predict than for cases (Fig 1A and 1B, as well as S2 and S3 Tables).

## Contribution to the forecast Hub

Of our three models, only the renewal model and the crowd forecast were included in the official Forecast Hub median ensemble ("hub-ensemble-realised"), while the convolution model was never included as it was deemed too similar to the existing renewal model. In the official Hub ensemble, there were on average 7.1 models included (including our own), with a median of 7, a minimum of 4 (on 28 December 2020 over the Christmas period) and a maximum of 10. Versions that included either all of our models ("hub-ensemble-with-all") or only one of them ("hub-ensemble-with-X") were computed retrospectively. An overview of all models and ensemble versions is shown in S10 Table.

For cases, our contributions (compared to the Hub ensemble without our contributions) consistently improved performance across all forecasting horizons (rel. WIS 0.9 two weeks

ahead, see S4 Table). Contributions from the crowd forecasts alone also improved performance of the Hub ensemble across all forecast horizons, while contributions from the renewal model had a negative effect for longer horizons (rel. WIS 1.02 three weeks ahead, 1.06 four weeks ahead). The realised ensemble including both models performed better or equal compared to all versions with only one model included for up to three weeks ahead, suggesting synergistic effects. Only for predictions four weeks ahead would removing the renewal model have improved performance (S5 Table). The realised ensemble performed comparably to the crowd forecasts for predictions one to two weeks ahead, and worse for greater forecast horizons.

For deaths, contributions from the renewal model and crowd forecast together improved performance only for one week ahead predictions and showed an increasingly negative impact on performance for longer horizons (rel. WIS of the Hub-ensemble-realised 1.01 two weeks ahead, 1.05 four weeks ahead, S4 and S5 Tables). Individual contributions from both the renewal model and the crowd forecast were largely negative, while a version of the Hub ensemble with only the convolution model included would have performed consistently better across all forecast horizons (with the positive impact increasing for longer horizons). This is especially interesting as the convolution model performed consistently worse than the pre-existing Hub ensemble (Fig 1) and especially worse for longer horizons.

We also considered the impact of our contributions on a version of the Hub ensemble constructed by taking the quantile-wise mean, rather than the median. General trends were similar, with the notable exception of the convolution model, which had a consistently positive impact on the median ensemble, but a mixed and mostly slightly negative impact on the mean ensemble (Fig 4B and S21(B) Fig). This may happen if a model is more correct directionally relative to the pre-existing ensemble, but overshoots in absolute terms, thereby moving the ensemble too far. For both the mean and the median ensemble, changes in performance from adding or removing models were of a similar order of magnitude, suggesting that at least in this instance, with a relatively small ensemble size, the median ensemble was not necessarily more 'robust' to changes than the mean ensemble. However, the ensemble version with all our forecasts included ("hub-ensemble-with-all") tended to perform relatively better for the median ensemble than the mean ensemble, suggesting that adding more models may be more beneficial or 'safer' for the median than for the mean ensemble as directional errors can more easily cancel out than errors in absolute terms.

## Discussion

Epidemiological forecasting modelling combines knowledge about infectious disease dynamics with the subjective opinion of the researcher who develops and refines the model. In this study, we compared forecasts of cases of and deaths from COVID-19 in Germany and Poland based purely on human judgement and elicited from a crowd of researchers and volunteers against forecasts from two semi-mechanistic epidemiological models. In spite of the small number of participants and a general tendency to be overconfident, crowd forecasts consistently outperformed our epidemiological models as well as the Hub ensemble when forecasting cases but not when forecasting deaths. This suggests that humans might be relatively good at foreseeing trends that are hard to model but may struggle to form an intuition for the exact relationship between cases and deaths.

Past studies have evaluated the performance of model-based forecasting approaches as well as human experts and non-experts in various contexts. However, most of these studies either focused only on the evaluation of (expert-tuned) model-based approaches [e.g. 12,13,14], or exclusively on human forecasts [19, 20, 24, 25]. In contrast, we directly compared human and

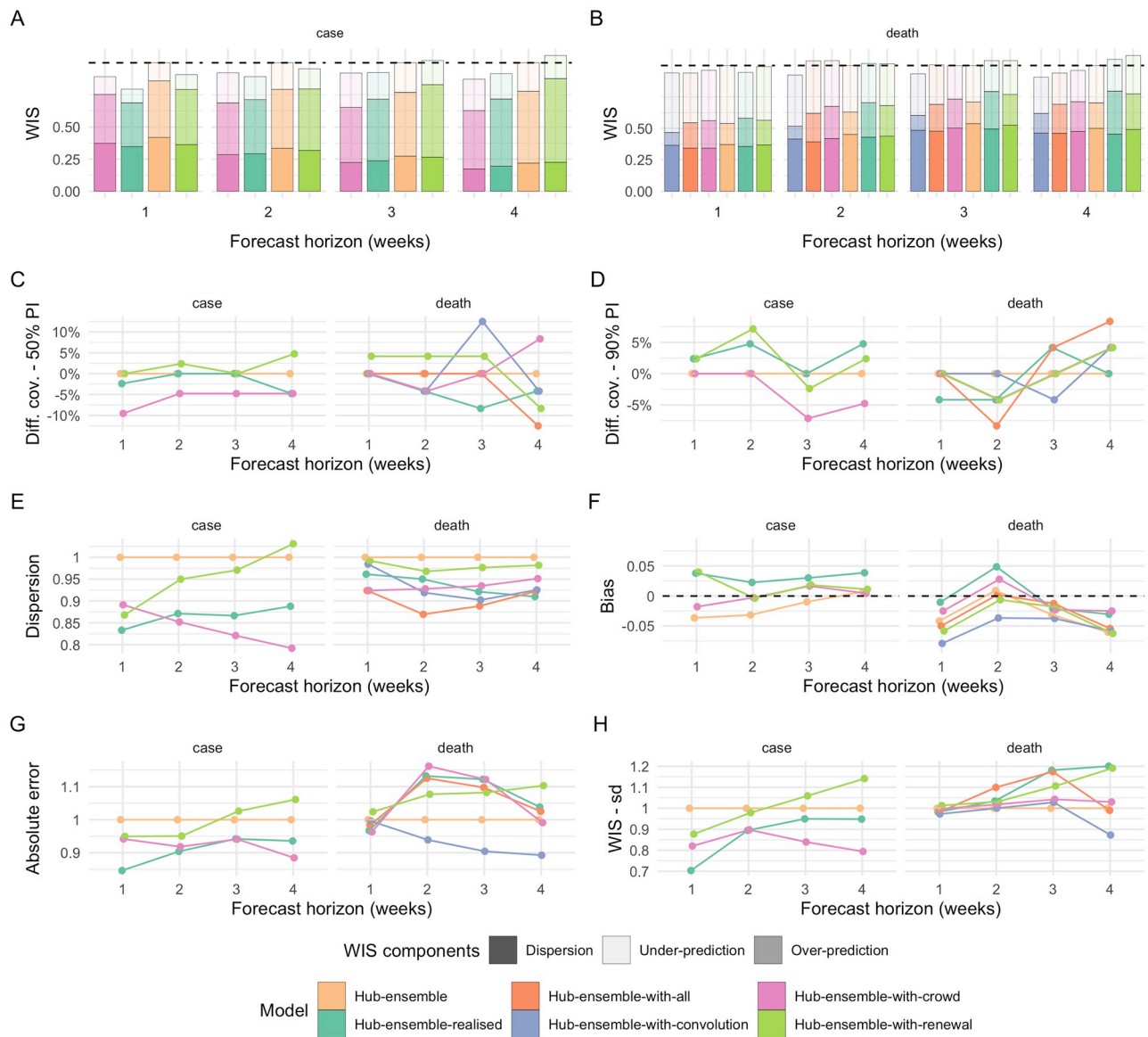

**Fig 4. Relative aggregate performance metrics across forecast horizons for different versions of the Hub median ensemble.** "Hub-ensemble" *excludes* all our models, Hub-ensemble-all *includes* all of our models, "Hub-ensemble-realised" is the actual hub-ensemble observed in reality, which includes the renewal model and the crowd forecasts, but not the convolution model. A, B: mean weighted interval score (WIS) across horizons relative to the Hub ensemble (lower values indicate better performance). C, D: Empirical coverage of the 50% and 90% prediction intervals minus empirical coverage observed for the Hub ensemble. E: Dispersion relative to the dispersion of the Hub ensemble. Higher values mean greater dispersion of the forecast and imply ceteris paribus a worse score. F: Bias, i.e. general (relative) tendency to over- or underpredict. Values are between -1 (complete under-prediction) and 1 (complete over-prediction) and 0 ideally. G: Absolute error of the median forecast relative to the Hub ensemble. H. Standard deviation of all WIS values for different horizons relative to the Hub ensemble.

model-based forecasts. This is similar to the approach taken by [11], but extends it in several ways. While Farrow et al. only asked for point predictions and constructed a predictive distribution from these, we asked participants to provide a full predictive distribution, allowing us to compare human forecasts and models without any further assumptions, as well as to analyse how humans quantified their uncertainty. In addition, we compared crowd forecasts to two semi-mechanistic models informed by basic epidemiological knowledge of COVID-19, allowing us to assess not only relative performance but also to analyse qualitative differences

 

between human judgement and model-based insight. In terms of interpretability of the results, exact knowledge of our two models, as well as focus on a limited set of targets and locations was a major advantage of our study compared to larger studies conducted by the Forecast Hubs [12–15, 17].

The strong performance of crowd forecasts in our study is in line with results from Farrow et al. who also report strong performance of human predictions in past Flu challenges despite difficulties to recruit a large number of participants. The advantage of crowd forecasts we observed over our semi-mechanistic models is likely in part explained by the fact that we compared an ensemble of crowd forecasts with single models. However, this probably explains only part of the difference, and performance relative to the Hub ensemble strongly suggests that human insight is valuable when forecasting highly volatile and potentially hard-to-predict quantities such as case numbers. One potential explanation is that humans can have access to data that is not available to or hard to integrate into model-based forecasts. Relatively good performance of our semi-mechanistic models short-term, but not longer-term, suggests that model-based forecasts are helpful to extrapolate from current conditions, but require some form of human intervention or additional assumptions to inform forecasts when conditions change over time. This human intervention may be particularly important when dealing with artefacts in reporting and data anomalies (and especially when using daily, rather than weekly data). The large variance in predictions in January in Germany for example (severe under-prediction followed by severe over-prediction, see Fig 3A), may in part be caused by the fact that the renewal model operated on daily data and therefore was susceptible to fluctuations in daily reporting which have less of an influence on weekly reporting.

Our results suggest that human intervention may be less beneficial when forecasting deaths (especially at shorter horizons, when deaths are largely dependent on already observed cases), which benefits from the ability to model the delays and exact epidemiological relationships between different leading and lagged indicators. Relatively good performance of the convolution model, especially compared to the poor performance of the renewal model on deaths (which used only deaths to estimate and predict the effective reproduction number) underlines the importance of including leading indicators such as cases as a predictor for deaths.

Given the low number of participants in our study, it is difficult to generalise conclusions about crowd predictions to other settings. Using R shiny as a platform for the web application arguably created some limits to user experience and performance, influencing the number of participants and potentially creating a self-selection effect. Motivating forecasters to contribute regularly proved challenging, especially given that the majority of our participants were from the UK and may not have been familiar with all relevant details of the situation in Germany and Poland. On the other hand, R shiny facilitated quick development and allowed us to provide our crowd forecasting tooling as an open source R package, meaning that it is available for others to use, for example in settings like early-stage outbreaks where model-based forecasts are not available. In light of the relatively small number of Hub ensemble models, performance of the Hub ensemble is also difficult to generalise. More research is needed to replicate these findings and investigate how crowd forecasts compare against the types of models and model ensembles policy makers use to inform their decisions.

Our work suggests that crowd forecasts and model-based forecasts could have different strengths and may be able to complement each other. When choosing a suitable approach for a given task it is important to take into account how the output will be used. In this work we focused on forecasts (which aim to predict future data points whilst accounting for all factors that might influence them), whereas policy makers might be more interested in projections (which show what would happen in the absence of any events that could change the trend) or scenario modelling. Forecasts may not be a suitable basis for informing policy decisions, if

forecasters already have factored in the expectation of a future intervention. Model-based approaches can be either forecasts or projections depending on the assumptions, whereas eliciting projections that are not influenced by implicit assumptions about the future from humans may be harder.

Further work should explore the effects of humans refining their mathematical models or changing model outputs in more detail. Model-based forecasts could be used as an input to human judgement, with researchers adjusting predictions generated by models. Seeing a model-based forecast could help humans calibrate uncertainty better, while allowing for manual intervention to adapt spurious trend predictions. Tools need to be developed to facilitate this process at a larger scale. Human insight could also be used as an input to models. Such a 'hybrid' forecasting approach could for example ask humans to predict the trend of the effective reproduction number $R_t$ or the doubling rate (i.e. how the epidemic evolves) into the future and use this to estimate the exact number of cases, hospitalisations or deaths this would imply. In light of severe overconfidence, yet good performance in terms of the absolute error, post-processing of human forecasts to adjust and widen prediction intervals may be another promising approach. Crowd forecasting in general could benefit greatly from the availability of tools suitable to appeal to a greater audience. Given the good performance we and previous authors observed in spite of the limited resources available and the small number of participants, this seems worthwhile to further develop and explore.

## Supporting information

**S1 Text. Further details on the semi-mechanistic forecasting models.**
(PDF)

**S1 Table. Overview of the scoring metrics used.**
(PDF)

**S2 Table. Scores for one and two week ahead forecasts.** Scores are cut to three significant digits and rounded). Note that scores for cases (which include the whole period from October 12th 2020 until March 1st 2021) and deaths (which include only forecasts from the 21st of December 2020 on) are computed on different subsets. Numbers in brackets show the metrics relative to the Hub ensemble (i.e. the median ensemble of all other models submitted to the German and Polish Forecast Hub, excluding our contributions). WIS is the mean weighted interval score (lower values are better), WIS—sd is the standard deviation of all scores achieved by a model. Dispersion, over-prediction and under-prediction together sum up to the weighted interval score. Bias (between -1 and 1, 0 is ideal) represents the general average tendency of a model to over- or underpredict. 50% and 90%-coverage are the percentage of observed values that fell within the 50% and 90% prediction intervals of a model.
(PDF)

**S3 Table. Scores for three and four weeks ahead forecasts.** Scores are cut to three significant digits and rounded). Note that scores for cases (which include the whole period from October 12th 2020 until March 1st 2021) and deaths (which include only forecasts from the 21st of December 2020 on) are computed on different subsets. Numbers in brackets show the metrics relative to the Hub ensemble (i.e. the median ensemble of all other models submitted to the German and Polish Forecast Hub, excluding our contributions). WIS is the mean weighted interval score (lower values are better), WIS—sd is the standard deviation of all scores achieved by a model. Dispersion, over-prediction and under-prediction together sum up to the weighted interval score. Bias (between -1 and 1, 0 is ideal) represents the general average

tendency of a model to over- or underpredict. 50% and 90%-coverage are the percentage of observed values that fell within the 50% and 90% prediction intervals of a model.
(PDF)

**S4 Table. Scores for one and two week ahead forecasts for the different versions of the median ensemble.** Scores are cut to three significant digits and rounded. Note that scores for cases (which include the whole period from October 12th 2020 until March 1st 2021) and deaths (which include only forecasts from the 21st of December 2020 on) are computed on different subsets. Numbers in brackets show the metrics relative to the Hub ensemble (i.e. the median ensemble of all other models submitted to the German and Polish Forecast Hub, excluding our contributions). WIS is the mean weighted interval score (lower values are better), WIS—sd is the standard deviation of all scores achieved by a model. Dispersion, over-prediction and under-prediction together sum up to the weighted interval score. Bias (between -1 and 1, 0 is ideal) represents the general average tendency of a model to over- or underpredict. 50% and 90%-coverage are the percentage of observed values that fell within the 50% and 90% prediction intervals of a model.
(PDF)

**S5 Table. Scores for three and four week ahead forecasts for the different versions of the median ensemble.** Scores are cut to three significant digits and rounded. Note that scores for cases (which include the whole period from October 12th 2020 until March 1st 2021) and deaths (which include only forecasts from the 21st of December 2020 on) are computed on different subsets. Numbers in brackets show the metrics relative to the Hub ensemble (i.e. the median ensemble of all other models submitted to the German and Polish Forecast Hub, excluding our contributions). WIS is the mean weighted interval score (lower values are better), WIS—sd is the standard deviation of all scores achieved by a model. Dispersion, over-prediction and under-prediction together sum up to the weighted interval score. Bias (between -1 and 1, 0 is ideal) represents the general average tendency of a model to over- or underpredict. 50% and 90%-coverage are the percentage of observed values that fell within the 50% and 90% prediction intervals of a model.
(PDF)

**S6 Table. Scores for one and two week ahead forecasts for the different versions of the mean ensemble.** Scores are cut to three significant digits and rounded. Note that scores for cases (which include the whole period from October 12th 2020 until March 1st 2021) and deaths (which include only forecasts from the 21st of December 2020 on) are computed on different subsets. Numbers in brackets show the metrics relative to the Hub mean ensemble (i.e. the mean ensemble of all other models submitted to the German and Polish Forecast Hub, excluding our contributions). WIS is the mean weighted interval score (lower values are better), WIS—sd is the standard deviation of all scores achieved by a model. Dispersion, over-prediction and under-prediction together sum up to the weighted interval score. Bias (between -1 and 1, 0 is ideal) represents the general average tendency of a model to over- or underpredict. 50% and 90%-coverage are the percentage of observed values that fell within the 50% and 90% prediction intervals of a model.
(PDF)

**S7 Table. Scores for three and four week ahead forecasts for the different versions of the mean ensemble.** Scores are cut to three significant digits and rounded. Note that scores for cases (which include the whole period from October 12th 2020 until March 1st 2021) and deaths (which include only forecasts from the 21st of December 2020 on) are computed on different subsets. Numbers in brackets show the metrics relative to the Hub mean ensemble (i.e.

the mean ensemble of all other models submitted to the German and Polish Forecast Hub, excluding our contributions). WIS is the mean weighted interval score (lower values are better), WIS—sd is the standard deviation of all scores achieved by a model. Dispersion, over-prediction and under-prediction together sum up to the weighted interval score. Bias (between -1 and 1, 0 is ideal) represents the general average tendency of a model to over- or underpredict. 50% and 90%-coverage are the percentage of observed values that fell within the 50% and 90% prediction intervals of a model.
(PDF)

**S8 Table. Scores for one and two week ahead forecasts (sensitivity analysis).** Scores are cut to three significant digits and rounded. In the original analysis, cases and deaths were scored on different periods, as the convolution model was only added later. This table shows performance of all models restricted to the period from December 14 2020 until March 1st 2021 where all models were available. Numbers in brackets show the metrics relative to the Hub ensemble (i.e. the median ensemble of all other models submitted to the German and Polish Forecast Hub, excluding our contributions). WIS is the mean weighted interval score (lower values are better), WIS—sd is the standard deviation of all scores achieved by a model. Dispersion, over-prediction and under-prediction together sum up to the weighted interval score. Bias (between -1 and 1, 0 is ideal) represents the general average tendency of a model to over- or underpredict. 50% and 90%-coverage are the percentage of observed values that fell within the 50% and 90% prediction intervals of a model.
(PDF)

**S9 Table. Scores for three and four week ahead forecasts (sensitivity analysis).** Scores are cut to three significant digits and rounded. In the original analysis, cases and deaths were scored on different periods, as the convolution model was only added later. This table shows performance of all models restricted to the period from December 14 2020 until March 1st 2021 where all models were available. Numbers in brackets show the metrics relative to the Hub ensemble (i.e. the median ensemble of all other models submitted to the German and Polish Forecast Hub, excluding our contributions). WIS is the mean weighted interval score (lower values are better), WIS—sd is the standard deviation of all scores achieved by a model. Dispersion, over-prediction and under-prediction together sum up to the weighted interval score. Bias (between -1 and 1, 0 is ideal) represents the general average tendency of a model to over- or underpredict. 50% and 90%-coverage are the percentage of observed values that fell within the 50% and 90% prediction intervals of a model.
(PDF)

**S10 Table. Overview of the models and ensembles used.**
(PDF)

**S1 Fig. Screenshot of the crowdforecasting app used to elicit predictions (made in June 2021).**
(TIF)

**S2 Fig. Visualisation of aggregate performance metrics for forecasts one to four weeks into the future in Germany.** A, B: mean weighted interval score (WIS, lower indicates better performance) across horizons. WIS is decomposed into its components dispersion, over-prediction and under-prediction. C: Empirical coverage of the 50% prediction intervals (50% coverage is perfect). D: Empirical coverage of the 90% prediction intervals. E: Dispersion (same as in panel A, B). Higher values mean greater dispersion of the forecast and imply ceteris paribus a worse score. F: Bias, i.e. general (relative) tendency to over- or underpredict. Values

are between -1 (complete under-prediction) and 1 (complete over-prediction) and 0 ideally. G: Absolute error of the median forecast (lower is better). H. Standard deviation of all WIS values for different horizons

(TIF)

**S3 Fig. Visualisation of aggregate performance metrics for forecasts one to four weeks into the future in Poland.** A, B: mean weighted interval score (WIS, lower indicates better performance) across horizons. WIS is decomposed into its components dispersion, over-prediction and under-prediction. C: Empirical coverage of the 50% prediction intervals (50% coverage is perfect). D: Empirical coverage of the 90% prediction intervals. E: Dispersion (same as in panel A, B). Higher values mean greater dispersion of the forecast and imply ceteris paribus a worse score. F: Bias, i.e. general (relative) tendency to over- or underpredict. Values are between -1 (complete under-prediction) and 1 (complete over-prediction) and 0 ideally. G: Absolute error of the median forecast (lower is better). H. Standard deviation of all WIS values for different horizons.

(TIF)

**S4 Fig. Visualisation of aggregate performance metrics across locations.** A, B: mean weighted interval score (WIS, lower indicates better performance) across horizons. WIS is decomposed into its components dispersion, over-prediction and under-prediction. C: Empirical coverage of the 50% prediction intervals (50% coverage is perfect). D: Empirical coverage of the 90% prediction intervals. E: Dispersion (same as in panel A, B). Higher values mean greater dispersion of the forecast and imply ceteris paribus a worse score. F: Bias, i.e. general (relative) tendency to over- or underpredict. Values are between -1 (complete under-prediction) and 1 (complete over-prediction) and 0 ideally. G: Absolute error of the median forecast (lower is better). H. Standard deviation of WIS values.

(TIF)

**S5 Fig. Visualisation of aggregate performance metrics across locations in relative terms.** A, B: mean weighted interval score (WIS) across locations (lower values indicate better performance). C, D: Empirical coverage of the 50% and 90% prediction intervals. E: Dispersion. Higher values mean greater dispersion of the forecast and imply ceteris paribus a worse score. F: Bias, i.e. general (relative) tendency to over- orunderpredict. Values are between -1 (complete under-prediction) and 1 (complete over-prediction) and 0 ideally. G: Absolute error of the median forecast. H. Standard deviation of WIS values.

(TIF)

**S6 Fig. Visualisation of daily report data.** The black line represents weekly data divided by seven. Data were last accessed through the German and Polish Forecast Hub on August 21 2021.

(TIF)

**S7 Fig. Visualisation of the absolute difference between the daily report data at the time and the data now.** In Germany, there were zero cases and deaths reported on 2020–10-12, and only later 2467 cases and 6 deaths were added. Data were last accessed through the German and Polish Forecast Hub on May 10 2022.

(TIF)

**S8 Fig. Visualisation of the relative difference between the weekly report data at the time and the data now.** Apart from the data that was retrospectively added on 2020–10-12, data updates did not have a noticeable effect on weekly data (as shown in the forecasting

application). Data were last accessed through the German and Polish Forecast Hub on May 10 2022.
(TIF)

**S9 Fig. Visualisation of forecasts and scores for one week ahead forecasts.** A, C: Visualisation of 50% prediction intervals of one week ahead forecasts against the reported values. Forecasts that were not scored (because there was no complete set of death forecasts available) are greyed out. B, D: Visualisation of corresponding WIS.
(TIF)

**S10 Fig. Visualisation of forecasts and scores for three week ahead forecasts.** A, C: Visualisation of 50% prediction intervals of three week ahead forecasts against the reported values. Forecasts that were not scored (because there was no complete set of death forecasts available) are greyed out. B, D: Visualisation of corresponding WIS.
(TIF)

**S11 Fig. Visualisation of forecasts and scores for three week ahead forecasts.** A, C: Visualisation of 50% prediction intervals of four week ahead forecasts against the reported values. Forecasts that were not scored (because there was no complete set of death forecasts available) are greyed out. B, D: Visualisation of corresponding WIS.
(TIF)

**S12 Fig. Distribution of weighted interval scores for one week ahead forecasts.** A: Distribution of weighted interval scores for one week ahead forecasts of the different models and forecast targets pooled across locations. B: Distribution of WIS separate by country.
(TIF)

**S13 Fig. Distribution of weighted interval scores for three week ahead forecasts.** A: Distribution of weighted interval scores for three week ahead forecasts of the different models and forecast targets pooled across locations. B: Distribution of WIS separate by country.
(TIF)

**S14 Fig. Distribution of weighted interval scores for four week ahead forecasts.** A: Distribution of weighted interval scores for four week ahead forecasts of the different models and forecast targets pooled across locations. B: Distribution of WIS separate by country.
(TIF)

**S15 Fig. Distribution of model ranks (in terms of WIS) for one week ahead forecasts.** A: Distribution of the ranks (determined by the weighted interval score) for one week ahead forecasts of the different models and forecast targets, pooled across locations. B: Distribution of ranks separate by country.
(TIF)

**S16 Fig. Distribution of model ranks (in terms of WIS) for two week ahead forecasts.** A: Distribution of the ranks (determined by the weighted interval score) for two week ahead forecasts of the different models and forecast targets, pooled across locations. B: Distribution of ranks separate by country.
(TIF)

**S17 Fig. Distribution of model ranks (in terms of WIS) for three week ahead forecasts.** A: Distribution of the ranks (determined by the weighted interval score) for three week ahead

forecasts of the different models and forecast targets, pooled across locations. B: Distribution of ranks separate by country.
(TIF)

**S18 Fig. Distribution of model ranks (in terms of WIS) for four week ahead forecasts.** A: Distribution of the ranks (determined by the weighted interval score) for four week ahead forecasts of the different models and forecast targets, pooled across locations. B: Distribution of ranks separate by country.
(TIF)

**S19 Fig. Difference in WIS between the Crowd forecast and the Hub ensemble for two week ahead forecasts.** Values below zero mean better performance of the Crowd forecasts.
(TIF)

**S20 Fig. Difference in WIS between the Crowd forecast and the Renewal model for two week ahead forecasts.** Values below zero mean better performance of the Crowd forecasts.
(TIF)

**S21 Fig. Visualisation of aggregate performance metrics across forecast horizons for the different versions of the Hub mean ensemble.** "Hub-ensemble" *excludes* all our models, Hub-ensemble-all *includes* all of our models, "Hub-ensemble-realised" is the actual hub-ensemble observed in reality, which includes the renewal model and the crowd forecasts, but ont the convolution model. Values (except for Bias) are computed as differences to the Hub ensemble which excludes our contributions. For Coverage, this is an absolute difference, for other metrics this is a percentage difference. A, B: mean weighted interval score (WIS) across horizons relative to the Hub ensemble (lower values indicate better performance). C, D: Empirical coverage of the 50% and 90% prediction intervals minus empirical coverage observed for the Hub ensemble. E: Dispersion relative to the dispersion of the Hub ensemble. Higher values mean greater dispersion of the forecast and imply ceteris paribus a worse score. F: Bias, i.e. general (relative) tendency to over- orunderpredict. Values are between -1 (complete under-prediction) and 1 (complete over-prediction) and 0 ideally. G: Absolute error of the median forecast relative to the Hub ensemble. H. Standard deviation of all WIS values for different horizons relative to the Hub ensemble.
(TIF)

**S22 Fig. Visualisation of aggregate performance metrics across forecast horizons (period from December 14th 2020 on).** From December 14th 2020 on, all models were available. In the original analysis, cases and deaths were scored on different periods, as the convolution model was only added later. This sensitivity analysis shows performance of all models restricted to the period from December 14 2020 until March 1st 2021 where all models were available. A, B: mean weighted interval score (WIS, lower indicates better performance) across horizons. WIS is decomposed into its components dispersion, over-prediction and under-prediction. C: Empirical coverage of the 50% prediction intervals (50% coverage is perfect). D: Empirical coverage of the 90% prediction intervals. E: Dispersion (same as in panel A, B). Higher values mean greater dispersion of the forecast and imply ceteris paribus a worse score. F: Bias, i.e. general (relative) tendency to over- or underpredict. Values are between -1 (complete under-prediction) and 1 (complete over-prediction) and 0 ideally. G: Absolute error of the median forecast (lower is better). H. Standard deviation of all WIS values for different horizons
(TIF)

**S23 Fig. Number of participants who submitted a forecast over time.**
(TIF)

**S24 Fig. Number of member models in the official Hub ensemble.** This includes our crowd forecasts and the renewal model. Note that the renewal model was not included in the ensemble on December 28th 2020.
(TIF)

**S25 Fig. Crowd forecasts and baseline shown in the application for a two week horizon.** Shown are the median, as well as the 50% and 90% prediction intervals (in order of decreasing opacity). For any given point in time, the baseline shown in red is what forecasters saw when they opened the app (the baseline shown was constant across all forecast horizons).
(TIF)

**S1 Acknowledgements. Members of the CMMID COVID-19 working group.**
(PDF)

## Acknowledgments

We thank all forecasters who participated in this study for their contribution.

## Author Contributions

**Conceptualization:** Nikos I. Bosse, Sam Abbott, Johannes Bracher, Edwin van Leeuwen, Anne Cori, Sebastian Funk.

**Data curation:** Nikos I. Bosse.

**Formal analysis:** Nikos I. Bosse, Sam Abbott.

**Investigation:** Nikos I. Bosse, Sam Abbott.

**Methodology:** Nikos I. Bosse, Sam Abbott.

**Software:** Nikos I. Bosse, Sam Abbott.

**Supervision:** Sam Abbott, Johannes Bracher, Edwin van Leeuwen, Anne Cori, Sebastian Funk.

**Visualization:** Nikos I. Bosse.

**Writing – original draft:** Nikos I. Bosse.

**Writing – review & editing:** Nikos I. Bosse, Sam Abbott, Johannes Bracher, Habakuk Hain, Billy J. Quilty, Mark Jit, Edwin van Leeuwen, Anne Cori, Sebastian Funk.

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
