## [Decision Letter · Decision Letter 0]

19 Apr 2022

Dear Mr Bosse,

Thank you very much for submitting your manuscript "Comparing human and model-based forecasts of COVID-19 in Germany and Poland" for consideration at PLOS Computational Biology. As with all papers reviewed by the journal, your manuscript was reviewed by members of the editorial board and by several independent reviewers. The reviewers appreciated the attention to an important topic. Based on the reviews, we are likely to accept this manuscript for publication, providing that you modify the manuscript according to the review recommendations.

There are some suggestions on how to improve the results (including minor adjustments to calculations) as well as requests for more detail on the approach to soliciting the human based forecasts. Please ensure these points are addressed. Furthermore, there are a number of more open-ended queries put to you that you should consider and address, likely in the Discussion. Doing so will strengthen the manuscript considerably by providing a stronger view on what this work adds, and where the logical next steps in research on human based forecasts lies. A considered extension to the Discussion (and other section of the manuscript as required) to explore these queries and make recommendations on next-steps and additional research on this important topic would be welcome.

Sincerely,

James M McCaw, PhD

Associate Editor

PLOS Computational Biology

Nina Fefferman

Deputy Editor

PLOS Computational Biology

[LINK]

Reviewer's Responses to Questions

**Comments to the Authors:**

Reviewer #1: Review is uploaded as an attachment.

Reviewer #2: # Overview

This study assessed the relative strengths and weaknesses of model-based and crowd-sourced COVID-19 forecasts produced for Germany and Poland, with the goal of assessing what a model can add beyond the subjective judgment of the modeller.

The authors found that crowd-sourced forecasts were over-confident, but outperformed model-based in predicting COVID-19 case numbers over a 1-4 week horizon.

In contrast, model-based forecasts outperformed the crowd-sourced forecasts in predicting COVID-19 deaths.

This provides a very timely and valuable perspective on model-based forecasts, and how these forecasts can complement forecasts that are explicitly opinion-based.

An interesting suggestion arising from this study is that "human insight may be most valuable when forecasting highly uncertain quantities [...] while mathematical models may be most useful in settings like predicting deaths, where leading indicators with a clear connection to the target variable are available."

# Major comments

1. Due to the limited number of participants for the crowd forecasts, and the minimal diversity of forecasting models, the greatest value of this study perhaps lies in demonstrating the merit of a larger comparison of crowd-based and model-based forecasts.

For reference, this study comprised only a small number of crowd forecasts.

There was a median of 6 participants for any given target, with only 2 participants — both authors of this study — submitting a forecast for every week.

Most participants were from the UK, not from Germany or Poland.

These crowd forecasts were compared to one model for case forecasts and to two models for death forecasts.

Of these two models, the convolution model (used only for death forecasts) was considered too similar to the renewal model to be included in the official Forecast Hub median ensemble.

Indeed, the authors acknowledge this limitation (page 21, lines 391-2: "Given the low number of participants in our study, it is difficult to generalize conclusions about crowd predictions to other settings.")

The challenges of recruiting participants for the crowd forecasts are indeed worth highlighting (§6, page 21).

While the authors state that their use of an R Shiny app "arguably created some limits to user experience and performance", the provided screenshot (Figure S1) suggests that they provided a very clear and simple interface for what is a rather challenging task!

Despite its limitations, this study provides interesting insights into the available forecasts, and how different forecasting methods may be best suited to predict different quantities.

It would be extremely interesting to see whether a larger study would support some of the observations reported here, such as model-based forecasts performing better for deaths than for cases (especially at longer forecast horizons) when compared to crowd forecasts.

2. Did the most recent case/death counts for a given forecasting week change in the data as reported in subsequent week(s)?

For example, in Figure 2C (Deaths in Poland) the crowd forecast around the start of December undershoots the reported values for a single week, but otherwise all prediction intervals around this time are in agreement with the reported values or overshoot them.

Similarly, the renewal model greatly undershoots some of the reported case counts (Figure 2A) and then jumps sharply upward the following week.

How much of this week-by-week change in bias might be due to reporting delays and incomplete data?

It would be great to see how the reported case and death numbers evolved over time (e.g., a time-lapse plot of the cases and deaths time-series in each country), to help the reader understand how the reported data may have affected the mechanistic models and the participants.

3. Is it fair to suggest that the performance of the crowd forecasts for case counts, relative to the renewal model, was perhaps driven as much by their overconfidence as it was by their smaller bias?

The crowd forecasts had smaller dispersion (Figure 1E) and fewer WIS outliers (Figures 3 and S8-S10), and the median forecasts had smaller absolute errors than the model-based forecasts (Figure 1G).

4. It would be great to see alternate versions of Figure 1 provided in the supplementary materials that present performance metrics separately for Germany and Poland.

This would help support some of the observations in the result sections, such as "relative to the Hub ensemble, the crowd forecasts performed noticeably better in Germany than in Poland and the renewal model better in Poland than in Germany" and "In general, there were fewer large outlier forecasts in Poland and in particular the renewal model performed more in line with other forecasts there."

Indeed, the renewal model appears to have performed substantially worse in Germany than in Poland (Figure 2A, 2B, and S5-S7).

After the November peak in Poland it appears that it may even have out-performed the crowd forecasts?

5. The authors make an interesting suggestion in §6:

"Human insight could also be used as an input to models.

Such a 'hybrid' forecasting approach could for example ask humans to predict the trend of the effective reproduction number Rt or the doubling rate (i.e. how the epidemic evolves) into the future and use this to estimate the exact number of cases, hospitalizations or deaths this would imply."

Indeed, an earlier remark about the German forecasts suggests that human insight may have played an important role in the crowd forecasts:

"[H]uman forecasters, possibly aware of the semi-lockdown announced on November 2nd 2020 (50) and the change in the testing regime (with stricter test criteria) on November 11th 2020 (36), were fastest to adapt to the new trend, and the Hub ensemble slowest."

Since 2 of the participants were authors of this study, are the authors able to comment whether they (and potentially other participants) were aware of these announcements when they submitted their predictions?

And how many participants submitted predictions in the relevant weeks of November?

# Minor comments

1. Figure 1: consider including zero in the y-axis scales for 1E, 1G, and 1H.

2. §5.3 (Death Forecasts, page 15): "The convolution model had a strong tendency to under-predict, which steadily decreased for longer forecast horizons."

The magnitude of the under-prediction decreased (i.e., the bias decreased but the value itself increased).

This could perhaps be worded more clearly, to avoid confusion about under-prediction growing larger for longer horizons.

3. Figure 3: what do the black squares and circles?

I suspect that they represent mean (circle) and median (square) values, but the figure legend and caption do not mention them.

4. Figure 4: consider adding an x-axis label (e.g., "Forecast horizon (weeks)"), and using consistent y-axis scales for sub-figures (e.g., the same scale for 4A and 4B, and same scales for case and deaths in each of 4C-4H).

5. Figure 4: I think that the caption contains several mistakes.

The panels look to be consistent with those in Figure 1, but the Figure 4 caption provides different descriptions for most panels.

For example, panels A and B appear to show mean WIS values, and panels C and D show empirical coverage of different intervals.

6. The number of models in the "Hub ensemble" (i.e., all models except those presented in this manuscript) is explained in §5.4 (page 18), but it could instead be mentioned in §4.4.

While reading through the result sections §5.1-5.3 I kept wondering how many models were in the ensemble.

7. The impact of including the convolution model in ensemble forecasts (§5.4) is indeed interesting!

I gather that the convolution model dragged down the "Hub-ensemble-with-convolution" forecasts, resulting in a larger negative bias than for the other ensembles (Figure 4F), but reducing the absolute error of the median forecast (Figure 4G).

Presumably the increased bias reflects outlying WIS values (as evident in Figures 3 and S8-S10)?

From Figures 2 and S5-S7, it seems plausible that the outlying WIS values for the convolution model were underestimates, primarily occurring in Germany over the month of January.

**Have the authors made all data and (if applicable) computational code underlying the findings in their manuscript fully available?**

Reviewer #1: Yes

Reviewer #2: Yes

PLOS authors have the option to publish the peer review history of their article (what does this mean?). If published, this will include your full peer review and any attached files.

Reviewer #1: No

Reviewer #2: No

Figure Files:

Data Requirements:

Reproducibility:

References:

---

## [Editor Report · Decision Letter 1]

18 Jul 2022

Dear Mr Bosse,

We are pleased to inform you that your manuscript 'Comparing human and model-based forecasts of COVID-19 in Germany and Poland' has been provisionally accepted for publication in PLOS Computational Biology.

Best regards,

James M McCaw, PhD

Associate Editor

PLOS Computational Biology

Nina Fefferman

Deputy Editor

PLOS Computational Biology

Jason A. Papin

Editor-in-Chief

PLOS Computational Biology

---

## [Editor Report · Acceptance letter]

1 Sep 2022

PCOMPBIOL-D-22-00086R1 

Comparing human and model-based forecasts of COVID-19 in Germany and Poland

Dear Dr Bosse,

I am pleased to inform you that your manuscript has been formally accepted for publication in PLOS Computational Biology. Your manuscript is now with our production department and you will be notified of the publication date in due course.

With kind regards,

Anita Estes
